# The Potential to Reduce Patient Co-Payment and the Public Payer Spending in Poland through an Optimised Implementation of the Generic Substitution: The Win-Win Scenario Suggested by the Real-World Big Data Analysis

**DOI:** 10.3390/pharmaceutics13081165

**Published:** 2021-07-28

**Authors:** Przemysław Kardas, Aneta Lichwierowicz, Filip Urbański, Beata Szadkowska-Opasiak, Ewa Karasiewicz, Paweł Lewek, Dominika Krupa, Marcin Czech

**Affiliations:** 1Department of Family Medicine, Medical University of Lodz, 90-136 Łódź, Poland; dr.beata@o2.pl (B.S.-O.); ewakarasiewicz@wp.pl (E.K.); pawel.lewek@umed.lodz.pl (P.L.); 2National Health Fund, 02-528 Warsaw, Poland; Aneta.Lichwierowicz@nfz.gov.pl (A.L.); filip.urbanski@nfz.gov.pl (F.U.); 3Department of Pharmacoeconomics, Institute of Mother and Child, 01-211 Warsaw, Poland; dominika.j.krupa@gmail.com (D.K.); marcin.czech@biznes.edu.pl (M.C.)

**Keywords:** generic substitution, generic drugs, drug costs, adherence, pharmacoepidemiology, Poland, retrospective studies, real-world data, big data

## Abstract

High medication costs are one of the major barriers to patient adherence. Medication affordability might be improved by generic substitution. The aim of this study was to assess the effectiveness of the implementation of generic substitution mechanisms in Poland. This was a retrospective analysis of nationwide real-world big data corresponding to dispensation of metformin preparations in 2019 in Poland. Relevant prescription and dispensation data were compared to assess the prevalence of generic substitution and its economic consequences. Among the 1,135,863 e-prescriptions analysed, a generic substitution was found in only 4.81% of the packs dispensed, based on e-prescriptions issued for metformin under its originator version and 2.73% under generic drugs. It is estimated that if these values were applied to the total Polish drug market, patients could lose the opportunity to lower their co-payment by 15.91% and the national payer to reduce its reimbursement expenditures by 8.31%. Our results point at the suboptimal implementation of generic substitution in Poland. Therefore, relevant actions need to be taken in order to maximise the benefits provided by this mechanism. It could not only lead to the win-win scenario in which both patients and the national payer are secured substantial savings, but it could also have a positive impact on patient adherence.

## 1. Introduction

Securing patient adherence to long-term therapies is one of the major challenges faced by modern public health. In its seminal report published in 2003, the WHO indicated that the level of non-adherence reached 50% in chronic treatments [1]. Although nearly two decades have passed, not much of an improvement may be observed in this field. New studies prove that non-adherence is still equally prevalent. For example, a recent meta-analysis found the prevalence of non-adherence to antihypertensive medications in Asia to be 48% [2]

As with all complex behaviours, non-adherence may have diverse underlying causative factors. The WHO model distinguishes five clusters of such drivers. Along with the patient, condition, medication, and healthcare system-related determinants, these are also social and economic factors which profoundly influence medication adherence [1]. Among these factors, drug costs play a very important role [3]. Higher out-of-pocket costs for patients have consistently been associated with various forms of non-adherence, including non-initiation, poor implementation (e.g., skipping or reducing doses), and early discontinuation of long-term therapies (poor persistence) [4].

Unfortunately, drug costs tend to grow, challenging the sustainability of public healthcare systems and creating serious obstacles to adherence. In such a case, at least a partial solution of this problem could be provided by a wider use of generic drugs. Current evidence supports this point, proving that more expensive drugs might be safely replaced by their more affordable generic equivalents [5].

In order to stimulate a wider use of generics, many countries allow for a mechanism of generic substitution. According to WHO, generic substitution is the practice of replacing a medicine, whether marketed under a trade or generic name, with a lower-priced alternative medicine (a branded or unbranded generic) [6]. Generic substitution is a widely used tool in the drug policy of healthcare systems. It secures higher savings for the healthcare system, intensifies competition between manufacturers, and increases the availability of treatment for patients. Particularly, it is the last of the mentioned advantages that plays an important role for many vulnerable groups, such as those suffering from multiple chronic conditions or the elderly, who often struggle with the overall burden of healthcare costs. 

For several European markets (Denmark, Finland, Greece, Spain, Netherlands, Ireland, Portugal, and Sweden), generic substitution is mandatory, while for others, it is just recommended (France, Norway, and Switzerland) [7]. Generic substitution was included in the official Drug Policy 2018–2020, a strategic document issued by the Polish government [8], in accordance with the WHO guidance, which advocated the use of generics to contain expenditure [9].

However, by providing more affordable therapies, generic substitution can offer benefits that go beyond cost containment. Most studies evaluating the use of generic (rather than brand-name) drugs, applied in the treatment of chronic diseases, show a significantly higher long-term adherence following treatment initiation [10]. For example, adherence to generic, versus brand-name, statins has been extensively studied. Recent studies proved adherence and persistence to be higher among generic statin recipients in Sweden [11] and Japan [12]. As compared to those initiating brand-name statins, patients initiating generic statins in the USA were more likely to adhere and had a lower rate of a composite clinical outcome (comprising of hospitalization for an acute coronary syndrome or stroke and all-cause mortality) [13]. Similar observations were made in other scenarios, e.g., among elderly patients receiving antidiabetics, in whom the substitution between branded and unbranded products (as well as between generics) of the same substance did not negatively affect adherence, not even in multiple switchers [14]. In French patients initiating bisphosphonates, the prescribing of a generic drug led to a higher persistence rate and to better implementation at 1 year [15]. Generic initiation was also associated with improved adherence to antidepressants [16], aromatase inhibitors [17], and imatinib [18]. 

Thus, a wider use of generic drugs seems to be reasonable, both from the perspective of the healthcare system (because of cost containment), as well as from the patient’s perspective, as an enabler of medication adherence and its positive clinical and economic consequences. There is evidence proving that a higher adherence is associated with a lower risk of hospitalisation and lower overall health care costs related to chronic conditions [19]. Thus, a greater use of generic therapies can reduce overall healthcare system expenditure, both directly and indirectly.

Among European countries, Poland was found to have higher rates of non-adherence. In a cross-European study assessing adherence to antihypertensive treatment, the average level of non-adherence was 44%, whereas in Poland this value was much higher, i.e., 58% [20]. Another study found non-adherence in Poland as high as 83.8% in selected chronic conditions [21]. Thus, the implementation of medication adherence-enhancing interventions (in particular, an effective use of generic substitution) is extremely important in Poland.

The Polish healthcare system, like many other European systems, is a health insurance system based on the principle of social solidarity. Health services are provided free of charge to those insured (i.e., practically the whole population) by both public and private healthcare providers, and their costs are covered by the only national health payer, i.e., the National Health Fund (NHF). The NHF also provides reimbursement for prescribed drugs. Nevertheless, most drugs are subject to an out-of-pocket co-payment by patients, which varies across and within drug classes. Several drugs of crucial importance for particular therapies are available at a lump sum of PLN 3.20 (PLN—Polish zloty; approximately PLN 4.50 = EUR 1, as of June 2021), and some are free of charge. In the case of other medicines, patients pay 30%, 50%, or 100% of total drug costs out-of-pocket, depending on the effectiveness of the drug, according to evidence-based criteria (e.g., homeopathic drugs are paid 100%). The co-payment is organized around the idea of stimulating the use of generic drugs, as a result of the adoption of the reference price system, based on ATC classes 5, 4, and 3. This system categorizes medicines that are considered interchangeable (e.g., an originator and its generic equivalents) into one cluster, enabling the public payer to cover the same reimbursement amount for all medicines included in that cluster. Consequently, originator drugs generate higher co-payments than generics. In these conditions, patients are financially incentivized to use generics, in order to lower their co-payments [6]. In order to optimize the cost of therapy to the patient, improve adherence, and generate savings for the public payer, current Polish legislation establishing the rules of the dispensation of reimbursed products in community pharmacies [22] makes it obligatory for pharmacists and technicians to offer a less expensive alternative, with an equivalent formulation, to patients filling a prescription for a reimbursed product. 

The effectiveness of the practical implementation of generic substitution has not been thoroughly studied in Poland yet. Considering the present high levels of co-payments for pharmacotherapy in Poland (reaching, on average, more than 60% of an original drug price in 2017) [23], it may be assumed that the full potential of generic substitution has not been reached. In this study, we hypothesized that currently, the mechanism of generic substitution is underused in Poland, leading to increased patient co-payments and public payer spending. Therefore, the overall aim of this study was to assess, based on real-world data for the general population, how effectively generic substitution is used in Poland, and to what extent this substitution can be optimised to increase the affordability of the drugs for patients. The study also analysed whether this potential optimisation could generate additional savings, rather than costs, for the national payer organisation (NHF). The analysis was possible only recently, due to the introduction of e-prescriptions, which, after being piloted in 2018, came into regular use in Poland in 2019. It was the first time that the data on issued prescriptions had been collected on a mass scale, as well as analysed and compared to the data on filled prescriptions. 

## 2. Materials and Methods

### 2.1. Data and Study Design

This was a retrospective, custom-made analysis of the 2019 anonymised drug prescription and dispensation data possessed by NHF. In our study, we adopted metformin as a model drug, which is widely used in diabetes care. The medication is not only available in Poland, in both a generic and an originator form, but it also plays a significant role in both patients’ co-payments and NHF’s reimbursement expenditures (in 2019, it accounted for 3.27% of the total patient co-payment, and 2.03% of the total NHF reimbursement budget, being number two on the NHF reimbursement list of drugs incurring the highest expenditures) [24].

The NHF database registers full information on prescription drugs dispensed in community pharmacies in Poland, regardless of whether a particular prescription was issued by a public or a private healthcare provider. This data includes the quality and quantity of the medicinal products dispensed, as well as the economic data (such as the cost of a particular drug), details on patient co-payments, and the reimbursement costs incurred by the NHF.

The prescription data came from e-prescriptions. After the pilot program was introduced in 2018, e-prescriptions started to be widely used in Poland in 2019, covering a large part of prescriptions. This allowed for the collection of the original data that was entered by practitioners on prescriptions when issuing them. The use of unique identifiers allowed the merging of original prescription and dispensation data in each individual case. Thus, it was possible to compare the medicinal products prescribed and dispensed, as well as to trace all the steps of generic substitution. 

Reimbursement and co-payment levels in Poland have changed over the time. In 2019, the reimbursement lists were amended six times. Following standard procedures, the currently binding reimbursement list has been used to calculate reimbursement as per the date of dispensation. A similar procedure has been used to calculate reimbursement as per the date of prescribing. It is noteworthy that due to the evolution of the national reimbursement lists, relevant values for the same medicinal product, as per prescribing and dispensation, may differ.

In this analysis, only single-compound drugs containing metformin (i.e., the ones matching the Anatomical Therapeutic Chemical (ATC) code of A10BA02) were analysed. Consequently, multi-compound drugs were not analysed. 

Various presentations of Glucophage (Merck Sante s.a.s., Lyon, France) were collectively counted as the ‘originator drug’, whereas all the other drugs containing metformin were collectively counted as ‘generic drugs’. In order to be considered an equivalent, a medicinal product was required to have the same strength and formulation (e.g., an immediate-release dosage or a modified-release dosage).

For the calculation of the volume of the total metformin market in Poland, we used the national dispensation data recorded in the NHF database. For the assessment of the generic substitution, only e-prescription data could be used. These represented approximately 10% of all prescriptions issued for metformin, as the remaining 90% were issued in traditional paper-based form, not allowing for the analysis of the drugs prescribed.

From a total number of 1,539,863 e-prescriptions issued in Poland for various metformin preparations in 2019, we excluded those which, for various reasons, were not reimbursed, and those which were not dispensed. In order to reflect the actual level of reimbursement and patient co-payment, we also excluded prescriptions dispensed with incorrect reimbursement applied. Thus, the final analysis included 1,135,863 e-prescriptions (see Figure 1 for details of the e-prescription selection).

### 2.2. Ethics

Analyses of aggregated, anonymised prescription and dispensation data does not involve ethical issues. Therefore, according to the policy of the Ethical Commission of the Medical University of Lodz, this analysis was not subject to the ethical approval procedure.

### 2.3. Statistical Analyses

In descriptive statistics, both the original numbers and percentage rates, calculated out of the total number of identified substitution cases, were presented, unless otherwise stated. Values were compared with relevant tests with a *p*-value of less than 0.05 considered significant.

## 3. Results

### 3.1. National Metformin Market

Details on the total metformin market in Poland in 2019 are presented in Table 1. According to this data, 10,973,123 prescriptions for metformin preparations were dispensed in Poland, based on which patients were dispensed 19,580,846 packs of various medicinal products containing metformin. As many as 74.55% of these prescriptions led to the dispensation of generic drugs, which altogether accounted for 70.06% of the total number of packs of the various medicinal products containing metformin being dispensed. 

The total cost of the dispensed metformin was 298,633,242 PLN, of which 192,183,292 (64.35%) was subject to reimbursements, whereas 106,449,949 (35.65%) was covered by patients as the out-of-pocket co-payments. The share of the generic drugs was 66.73% in total costs, 76.13% in reimbursements, and 49.78% in co-payments. On average, the cost of one single-pack of dispensed generic metformin was 14.53 PLN, out of which patients paid 3.86 PLN on an out-of-pocket basis (26.59% of the original total price). The cost of one pack of the dispensed originator drug was 16.94 PLN on average, out of which the out-of-pocket co-payment made by patients amounted to 9.12 PLN (53.82% of the original total price). 

### 3.2. Generic Substitution

The analysed group of e-prescriptions ultimately included 1,135,863 prescriptions for various metformin formulations (see Figure 1). Out of this number, only 4.08% (46,295) were dispensed with substitution. For packs dispensed, the relevant percentage was even lower, i.e., 3.36% (70,064 out of a total of 2,085,954, see Table 2). Among specific age groups, substitution occurred most frequently among patients aged 18–29 years (4.11% out of the total number of packs dispensed in this age group, *p* < 0.01). The percentage of drug packs dispensed with substitution was very similar across genders (females: 3.38%, males: 3.33%, *p* > 0.05). Finally, there was a significant difference in the percentages of drug packs dispensed with substitution between drugs originally prescribed as the originator and a generic drug (4.81% vs. 2.73%, *p* < 0.01), although both those numbers were low.

### 3.3. Economic Consequences of Generic Substitution

Dispensation of metformin e-prescriptions, issued with and without substitution, was analysed in terms of its economic consequences (Table 3). Dispensation with generic substitution enabled the patient to save, on average, PLN 0.99 per pack of a metformin preparation. This amount could be nearly doubled, reaching PLN 1.78, if the substitution was optimised, i.e., if in each case the preparation with the lowest possible co-payment level was dispensed, respectively. Interestingly, substitution also ensured savings to the payer since, as a result of the substitution, the NHF paid, on average, PLN 0.28 less for reimbursement of one pack of a metformin preparation. Here again, the optimisation of reimbursements could increase NHF savings (in fact, tripling it (increasing the amount to PLN 0.94)).

Dispensation without generic substitution resulted in lost savings for both the patient and NHF (on average, PLN 1.04, and PLN 1.18, respectively, per one pack of a metformin preparation dispensed).

### 3.4. Lost Savings Due to Suboptimal Generic Substitution

We used the values of the lost savings to calculate the total loss for both patients and the NHF in Poland in 2019, due to dispensation of various metformin preparations without economic optimisation, i.e., savings that would be potentially possible with generic substitution. Table 4 presents the results of those calculations, for various levels of optimisation, i.e., various percentages of metformin preparations dispensed in the form of generics and the originator. For example, the optimal substitution of just 5% of packs of metformin originally dispensed as the originator would save the patients PLN 304,881, while saving the payer PLN 345,922. In case of drugs dispensed as generics, optimal substitution would generate a savings of PLN 541,851 and PLN 452,686 for the patients and NHF, respectively. Reaching the level of 100% optimisation would save the patients PLN 10,837,030 on generics and PLN 6,097,610 on originator drugs, making the total savings PLN 16,934,640. Relevant numbers for NHF would be similar, i.e., PLN 9,053,721 saved on generics, and PLN 6,918,443 saved on the originator, with a total of PLN 15,972,164 saved on the whole metformin market.

The value of the total potential savings achievable, by means of optimised generic substitution, was compared to the whole metformin market in Poland in 2019. Figure 2 shows the percentage of savings that could be made, for both the patients and the payer, with the optimal use of generic substitution reaching 15.91% of total patient co-payments and 8.31% of total reimbursements incurred by NHF on prescriptions issued for metformin, respectively.

If the same proportion of achievable savings was compared to the total patient co-payment budget in 2019, which amounted to 3.253 billion PLN, potential savings for the patients could exceed 518 million PLN. In 2019, the NHF spent 9.455 billion PLN on the reimbursement of prescription drugs; therefore, potential savings, due to optimised substitution, could be estimated at 786 million PLN.

## 4. Discussion

To the authors’ knowledge, this is the first large, nationwide, population-based study on the effectiveness of the implementation of the mechanism of generic substitution in Poland and one of very few, such wide-scale studies worldwide. Using real-world big data, we found a low prevalence of generic substitution applied to only 4.81% of packs dispensed, based on metformin e-prescriptions issued for its originator version. Another interesting observation was the even less frequent use of the same mechanism with e-prescriptions issued for various preparations of generic metformin, which were the subject of the further generic substitution in only 2.73% of dispensed packs, despite the potential savings the patients could obtain.

This suboptimal use of generic substitution was a reason the patients were losing large savings. Extrapolating our results to the total national metformin market in 2019, those lost savings were equivalent to nearly 16% of the total amount that the patients spent on the co-payments for various metformin preparations. In other words, the patients lost the chance of reducing their out-of-pocket co-payments by one eighth and to help their long-term adherence at the same time.

However, perhaps our most interesting finding was the fact that the patients’ lost savings were parallel to the lost savings of the national payer, i.e., the NHF. Due to suboptimal use of generic substitution, the NHF lost a chance to save over 8% of the money spent on reimbursement of metformin preparations.

Thus, our findings prove that the optimised generic substitution of metformin could lead to a win-win scenario, i.e., along with increased affordability of drugs and its positive impact on patient adherence, it could lead to substantial savings for the national payer. If these findings are extended to the total Polish drug market, the potential savings achieved by both patients and the payer may be hundreds of millions PLN.

A study performed in 17 low-and middle-income countries showed that, on average, 60% (range: 9–89%) could be saved by an individual country from a switch in the private sector purchases from originator brands to the lowest-priced generics [25]. This strong financial incentive is turning many such countries, e.g., China, to the active promotion of the use of generics [26]. Nonetheless, high-income countries are also able to benefit from the use of generic substitution. For example, changes introduced in Greece (since 2010) aiming at the reduction of public pharmaceutical expenditure which, among others, included generic substitution, proved effective. The average price per package declined in 2013 by 28%, from EUR 17.8 in 2012 to EUR 12.8 in 2013 [27]. Similarly, in Ireland, claimants’ costs were reduced by one-third when patients were changed to an equivalent cheaper, or generic, brand of proton pump inhibitor (PPI), while continuing on their original dose and quantity [28]. Therefore, European countries were advised to adopt various available measures to increase the use of generics, as a critical cost containment measure [29]. This approach was also reflected in the guidelines developed by the American College of Physicians, according to which clinicians are recommended to prescribe generic medications, if possible, rather than more expensive brand-name medications [30]. 

This, however, does not close the list of benefits provided by generic medicines, which offer much more to society than just their cost-saving potential through reduced prices. Apart from their cost-saving potential, generic medicines have an additional societal value by providing an easier access to pharmacotherapy, a stimulus for the innovation of pharmaceutical companies, and, last but not least, helping medication adherence [31]. A clear effect of the relationship between medication affordability and adherence was demonstrated in Catalonia, where the introduction of a fixed co-payment was followed by a statistically significant increase in initial medication non-adherence, which was reversed after the suspension of the fixed co-payment [32]. Similarly, in the USA, federal and state generic drug policies lowering cost-sharing were associated with an increase in patient’s medication use and adherence [33]. 

Aiming to increase the relative consumption of generics and generic substitution, some countries adopted various interventions, such as prescriptions by an international non-proprietary name (INN) of an active ingredient. This ensures that the choice of a specific brand is based, to a lesser extent, on marketing and behavioural factors, and more on economic calculations. INN prescribing is mandatory in several European countries (for example, in Estonia, France, Greece, Spain, Netherlands, Portugal, Slovakia, and Italy) and outside Europe (Australia) [8]. Other interventions include increasing confidence in generics and promoting their acceptance by professionals, patients, and the general community, as well as incentivizing pharmacists and physicians to prescribe generics more frequently [25]. Such solutions are applied, for example, in France and Hungary. Specifically, in France, pharmacies receive bonuses for high rates of generic dispensation [34].

In order to further increase the use of generics, it is advisable to employ policies intended to affect prescribing behaviours among physicians, such as guidelines, information (about prices and less expensive alternatives), and feedback [35]. This direction is particularly worth exploring in Poland, as a survey conducted among physicians proved that many of them doubted the equivalence of generic and brand name drugs, which prevented them from prescribing less expensive drugs [36]. In the light of our findings, which prove that 30% of metformin prescriptions issued in Poland in 2019 were prescribed for the originator, this problem seems to remain unsolved.

Therefore, along with stimulating generic prescribing, another approach seems to be advisable, i.e., a wider use of generic substitution at the level of community pharmacies. The currently binding Polish legislation obliges community pharmacists to inform patients on the availability of less expensive drug equivalents whenever such equivalents are available. It is noteworthy that in the case of common drugs such as metformin, such availability is a rule. Our results prove, however, that this legislation is not effective in securing the optimal implementation of the substitution; therefore, the question is how to improve this situation. Perhaps, one of the reasons is that pharmacists are not always aware of this obligation. A survey showed that one in five pharmacists (20.7%) did not know that each pharmacy had to inform patients about their option to replace a drug they had been prescribed with its less expensive equivalent [37]. Additionally, this obligation extends only to providing the patient with information of the option to obtain a substitute; however, there is no formal enforcement mechanism. Patients are free to purchase the brand of their choice, regardless of the availability of less expensive options. In such a case, patients’ beliefs and opinions play an important role. These, however, are not necessarily supportive for generics. Although current evidence does not prove inferiority of generics, as compared to brand-name drugs [38], the existence of misconceptions may even lead to side effects and worse outcomes, due to the so-called nocebo effect [39]. Moreover, Polish patients tend to overestimate the choice made by the prescriber who issues a prescription (who is not necessarily fully aware of availability of less expensive drug equivalents). This, however, may discourage patients from accepting generic substitution offered in a community pharmacy [40]. As a result, substitution utilization varies, depending on the level of awareness of the pharmacist and the patient, as well as an availability of certain products.

In fact, even the mandatory generic substitution does not offer a complete solution of the problem. For example, the introduction of mandatory generic substitution in South Africa showed diverse effects of the use of generics and originators among four studied groups of drugs. After the implementation of the law, generic SSRIs replaced originator products, and the effect on ACE-I and calcium channel blockers was less pronounced; in case of PPIs, the intended effect of the policy was not observed [41].

Perhaps, further studies are required to establish which methods of optimisation of generic substitution implementation could work best in Poland. Nevertheless, our findings clearly prove that this direction is worth following, both due to the potential patients’ savings for the patient and the payer, as well as the added value of improved medication adherence. Therefore, Poland should respond positively to the call, which urged governments to act appropriately and implement a coherent set of policies to increase the use of generic medicines [31].

There are several limitations of this study, which need to be considered. Firstly, metformin is an important drug for public health, because of its basic role in diabetes care. It accounts for a substantial portion of drug expenditures incurred both by patients and the national payer. Nevertheless, calculations based on this drug alone may not exactly reflect the overall tendencies in generic substitution, or its economic consequences in Poland. Secondly, in order to trace the pathways of substitution, we needed to limit our analysis to e-prescriptions, which constituted only one-tenth of the total volume of national prescriptions in 2019. Finally, our calculations of potential savings are based on the assumption that the least expensive generic is fully available, which is not always the case. This all could be a source of the potential bias of our model.

On the other hand, our analysis is the first one of this kind in Poland. Based on a large portion of the nationwide dispensation database, we were provided a unique opportunity to assess the extent and the economic consequences of generic substitution in real-world big data. A particular strength of our study comes with the use of methodology which gave grounds for the detailed assessment of the economic consequences of each individual episode of dispensing, regardless of whether it was associated with generic substitution or not, despite frequent changes of the reimbursement lists effective in the observed period. 

Thus, we are convinced that this study provides new, important information, which can stimulate optimisation of the generic substitution implementation in Poland. Currently available national drug policy accepted the horizon of 2022 [8]; therefore, in its new version, these issues are undoubtedly worth tackling. Our findings prove that an optimised use of generic substitution can generate substantial savings to both patients and the national payer. Owing to better drug affordability, it may lead to an improved adherence, as well. We also hope that the win-win scenario, which we have identified, is a strong incentive that will stimulate further research on generic substitution in Poland and abroad, in a search for the best practices of optimisation of this process. 

## 5. Conclusions

This study was the first large, nationwide, population-based study on the effectiveness of the implementation of generic substitution in Poland. Real-world big data confirmed the low prevalence of generic substitution, applied to only 4.81% of packs dispensed, based on e-metformin prescriptions issued for its originator version (and with only 2.73% of those prescribed as various generic drugs). This suboptimal use of generic substitution was a reason for large savings lost by both the patients and the national payer. Thus, our findings indicate the need to optimize the implementation of generic substitution in Poland. It could not only lead to a win-win scenario, where both the patients and the national payer are secured substantial savings, but it could also have a positive impact on patient adherence.

## Figures and Tables

**Figure 1 pharmaceutics-13-01165-f001:**
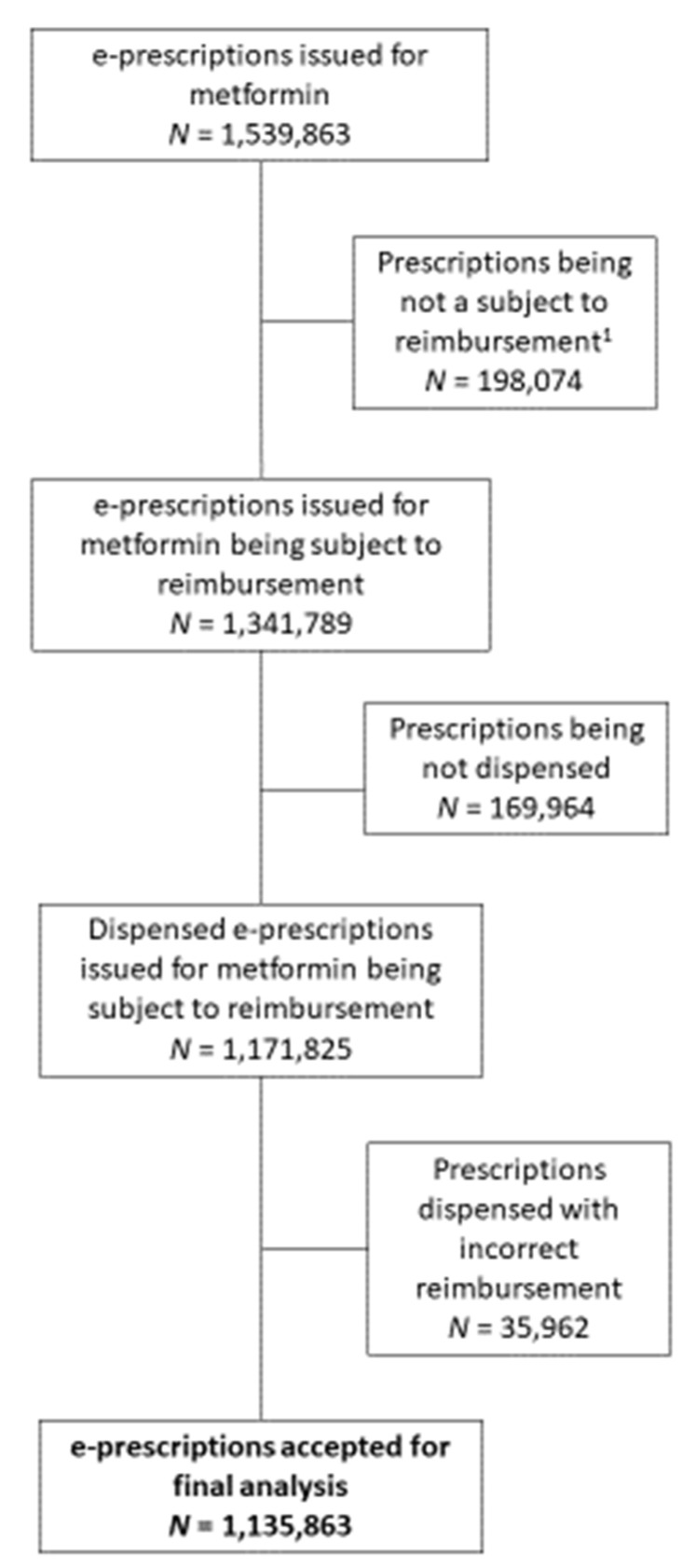
Flow chart presenting a selection of e-prescriptions issued for metformin in Poland in 2019 to be included in the final analysis; (1) e-prescriptions issued to citizens who did not benefit from reimbursement, e.g., those not covered by NHF insurance, foreigners, etc.

**Figure 2 pharmaceutics-13-01165-f002:**
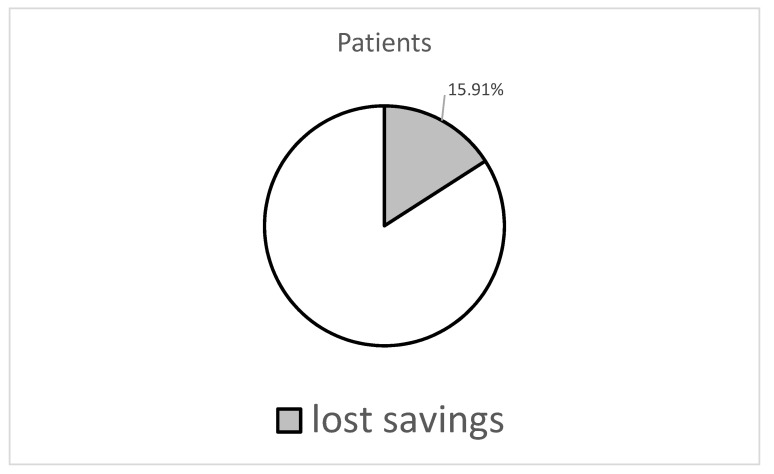
Potential savings that could be achieved by both the patients and the payer due to optimal use of generic substitution, expressed as the percentage of total co-payment (patients) and the percentage of total reimbursement (NHF), corresponding to all metformin prescriptions dispensed in 2019 in Poland. NHF—National Health Fund.

**Table 1 pharmaceutics-13-01165-t001:** Metformin market in Poland in 2019; data refer to the dispensation period between 1 January–31 December 2019; # numbers and percentages do not sum up to the total, as some prescriptions being issued for more than one pack of drug were dispensed in a form of both a generic and the originator drug. NHF: National Health Fund. PLN: Polish zlotys.

Drug	Prescriptions #		Packs		Total Drug Costs		Reimbursement Incurred by NHF		Patient Co-Payment	
	*N*	%	*N*	%	PLN	%	PLN	%	PLN	%
Generics	8,180,452	74.55	13,717,759	70.06	199,289,203	66.73	146,303,153	76.13	52,986,049	49.78
Originator	2,881,556	26.26	5,863,087	29.94	99,344,039	33.27	45,880,139	23.87	53,463,900	50.22
Total	10,973,123	100.00	19,580,846	100.00	298,633,242	100.00	192,183,292	100.00	106,449,949	100.00

**Table 2 pharmaceutics-13-01165-t002:** The analysis of dispensation of metformin e-prescriptions issued in Poland in 2019 (data refer to the dispensation period between 1 January–31 December 2019); # other forms of dispensation which do not satisfy the definition of generic substitution, such as dispensation of a pack with a dosage or a number of units other than those prescribed, other formulation (e.g., an immediate-release formulation instead of a modified-release one), etc.

Variable		Dispensed without Generic Substitution		Dispensed with Generic Substitution		Other #		Total
		Number of Packs Dispensed	%	Number of Packs Dispensed	%	Number of Packs Dispensed	%	
Age	0–17	1621	85.36	64	3.37	214	11.27	1899
	18–29	10,992	82.04	551	4.11	1855	13.85	13,398
	30–49	117,348	82.31	5627	3.95	19,596	13.74	142,571
	50–69	831,280	83.60	35,275	3.55	127,800	12.85	994,356
	70–89	742,650	81.62	27,753	3.05	139,449	15.33	909,853
	90+	19,241	80.58	794	3.33	3843	16.09	23,878
Gender	Female	951,449	82.41	39,050	3.38	164,022	14.21	1,154,520
	Male	771,685	82.85	31,014	3.33	128,735	13.82	931,434
Drug prescribed	Generic	1,202,735	82.45	39,881	2.73	216,175	14.82	1,458,792
	Originator	520,398	82.98	30,183	4.81	76,581	12.21	627,162
Total		1,723,133	82.61	70,064	3.36	292,757	14.03	2,085,954

**Table 3 pharmaceutics-13-01165-t003:** Economic consequences of dispensing e-prescriptions for metformin preparations with and without generic substitution in Poland in 2019 (data refer to the dispensation period between 1 January–31 December 2019); # non-zero values resulting from the change of the reimbursement level between the date of prescribing and dispensation; NHF—National Health Fund; PLN—Polish zlotys.

Payer	Type of Savings (Method of Calculation)	Dispensed with Generic Substitution		Dispensed without Generic Substitution	
		Total Savings (PLN)	Average Savings per 1 Pack Dispensed (*N* = 70,064)	Total Savings (PLN)	Average Savings per 1 Pack Dispensed (*N* = 1,723,133)
Patient	Real savings (co-payment as per prescription—co-payment paid)	69,640	0.99	−1025 #	0.00
	Maximal potential saving (co-payment as per prescription—minimal co-payment)	124,797	1.78	1,799,870	1.04
	Lost saving (co-payment paid—minimal co-payment)	55,633	0.79	1,800,895	1.04
NHF	Real savings (reimbursement as per prescription—reimbursement incurred)	19,743	0.28	1267 #	0.00
	Maximal potential saving (reimbursement as per prescription—minimal reimbursement)	65,821	0.94	2,028,644	1.18
	Lost saving (reimbursement paid—minimal reimbursement)	46,324	0.66	2,027,465	1.18

**Table 4 pharmaceutics-13-01165-t004:** Lost savings due to suboptimal implementation of generic substitution of metformin preparations prescribed and dispensed in Poland in 2019 (data refer to the dispensation period between 1 January–31 December 2019); NHF—National Health Fund; PLN—Polish zlotys.

Parameter		Metformin Preparations Dispensed in the Form of Generics (*N* = 13,717,759 Packs)		Metformin Preparations Dispensed in the Form of the Originator (*N* = 5,863,087 Packs)		Whole Metformin Market (*N* = 19,580,846 Packs)	
		Savings Lost by the Patient (PLN)	Savings Lost by NHF (PLN)	Savings Lost by the Patient (PLN)	Savings Lost by NHF (PLN)	Savings Lost by the Patient (PLN)	Savings Lost by NHF (PLN)
Optimisation premium (PLN/1 pack)		0.79	0.66	1.04	1.18	x	x
Optimisation level (%)	5	541,851	452,686	304,881	345,922	846,732	798,608
	10	1,083,703	905,372	609,761	691,844	1,693,464	1,597,216
	20	2,167,406	1,810,744	1,219,522	1,383,689	3,386,928	3,194,433
	30	3,251,109	2,716,116	1,829,283	2,075,533	5,080,392	4,791,649
	50	5,418,515	4,526,860	3,048,805	3,459,221	8,467,320	7,986,082
	75	8,127,772	6,790,291	4,573,208	5,188,832	12,700,980	11,979,123
	90	9,753,327	8,148,349	5,487,849	6,226,598	15,241,176	14,374,947
	100	10,837,030	9,053,721	6,097,610	6,918,443	16,934,640	15,972,164

## Data Availability

The data that support the study findings are made available by the authors with the permission of NHF (data owner). Restrictions apply to the availability of the data which were used under the license for this study.

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
