# Peer review of "The Potential to Reduce Patient Co-Payment and the Public Payer Spending in Poland through an Optimised Implementation of the Generic Substitution: The Win-Win Scenario Suggested by the Real-World Big Data Analysis"

_pharmaceutics, 2021, doi:10.3390/pharmaceutics13081165_

Round 1
Reviewer 1 Report
The topic of the study is interesting since it is related to generic substitution. The authors use data 2019 in Poland to make projections from hypotheses corresponding to different rates of substitution. The principal result is that in 2019 and related to metformin medication, the substitution rate is quite low 4.81%.
I’m not sure this study brings the results the authors suggest in the title, at least an inventory of generic substitution in Poland for one medication in 2019. There is no comparison with previous years to analyse the impact of the novel measures taken in Poland. Overall, the manuscript is well written ,particularly the introduction and discussion sections which are very informative and highly referecend. Nevertheless, the manuscript suffers from substantial inacurracies and overinterpretation. Moreover, the fact that the data are issued uniquely form Poland led to a loss of generalisability since its related to the polish healhcare system and its specific rules. At least, this manuscript allows to add a supplementary proof in a specific country in Europe of the potential benefit of generic substitution on cost savings.
Moreover, the interpretation related to the improvement of medication adherence is not supported by the results of the study but rather an interpretation of the authors of the potential consequences of a better coverage of generic substitution. The authors should delete this reference to the improvement of medication adherence in their title.
Concerning the method section : The methodology employed to calculate potential savings as reported in table 3 and 4 is hardly understandable, what is the lowest possible co-payment level (line 241) ? is it possible to give the formula employed to produce these results ?
We understand that the authors have made distinct hyptoheses which are explained in the results section but not in the methodology one.
The authors should define more the concepts they use : please define cost saving
Some of the results concerning the flow chart are presented in the method section and not in the flow chart.
In the results section :
I don’t understand the gap between the number of e-prescription presented in the flow chart and the number of 10 973 213 prescriptions for metformin. Does it mean that in this analysis only 10% of prescriptions of metformine are studied ? Please clarify this in the method section.
In paragraph 2, on generic substitution, it is not clear for the gap between the 1 135 863 prescriptions included in the analysis and the overall number of packs 2 085 954, the title of the table 2 is misleading since it refer to the dispensation and not the packs themselves.
To my opinion, fig 3 is not necessary.
Discussion
I m not sure that this study can be considered as a study on effectiveness of generic substitution in Poland
Author Response
Dear Reviewer No. 1,
Thank you for your thorough review. Here we come with clear response to the points raised by you, as well as some additional explanations.
- I’m not sure this study brings the results the authors suggest in the title, at least an inventory of generic substitution in Poland for one medication in 2019. There is no comparison with previous years to analyse the impact of the novel measures taken in Poland. Overall, the manuscript is well written ,particularly the introduction and discussion sections which are very informative and highly referecend. Nevertheless, the manuscript suffers from substantial inacurracies and overinterpretation. Moreover, the fact that the data are issued uniquely form Poland led to a loss of generalisability since its related to the polish healhcare system and its specific rules. At least, this manuscript allows to add a supplementary proof in a specific country in Europe of the potential benefit of generic substitution on cost savings.
RE: In order to better reflect the results collected, we have modified the title by removing ‘adherence’ from it. Our analysis could cover year 2019 only, as this was the year of nationwide introduction of e-prescriptions, allowing for the comparison of prescriptions issued, and dispensed for the first time in Poland. However, the methods of stimulation of generis substitution have not changed in Poland during last year, and are stabile for more than a decade. In such a sense, there was no ‘novel measure’ to test.
- Moreover, the interpretation related to the improvement of medication adherence is not supported by the results of the study but rather an interpretation of the authors of the potential consequences of a better coverage of generic substitution. The authors should delete this reference to the improvement of medication adherence in their title.
RE: So we did, see the above.
- Concerning the method section : The methodology employed to calculate potential savings as reported in table 3 and 4 is hardly understandable, what is the lowest possible co-payment level (line 241) ? is it possible to give the formula employed to produce these results ?
RE: Unfortunately, there is no secret formula to calculate this. For each time period, the lowest applicable co-payment, and reimbursement level, have to be find out on the list of reimbursed drugs. Note that this list is changing every 3 months, according to the update issued by the Ministry of health.
- We understand that the authors have made distinct hyptoheses which are explained in the results section but not in the methodology one.
RE: Our hypothesis was provided in more explicit way in the last paragraph of the introduction.
- The authors should define more the concepts they use : please define cost saving
RE: We believe that ‘cost saving’ is a term which is too general to be used in this context. Therefore, In Table 3 we defined various parameters which are more applicable to the various dimensions of our analysis..
- Some of the results concerning the flow chart are presented in the method section and not in the flow chart.
RE: see our answer to the next point
- In the results section : I don’t understand the gap between the number of e-prescription presented in the flow chart and the number of 10 973 213 prescriptions for metformin. Does it mean that in this analysis only 10% of prescriptions of metformine are studied ? Please clarify this in the method section.
RE: traditional paper-based prescriptions were not covered by our analysis, because neither their number, nor the details of prescribed drugs were available. We could only analyze e-prescriptions, in which case, both the details of prescribed, as well as dispensed prescriptions were available for the comparisons. All these details are made available in Figure 1. To make this more transparent to the readers, we added some additional explanations to the Methods section. Finally, the proportion of analyzed prescriptions was close to 10% of the total, as the relevant numbers were: 10 973 123 prescriptions for Metformin preparations dispensed in total, of which 1 171 825 were e-prescriptions subject to reimbursement.
- In paragraph 2, on generic substitution, it is not clear for the gap between the 1 135 863 prescriptions included in the analysis and the overall number of packs 2 085 954, the title of the table 2 is misleading since it refers to the dispensation and not the packs themselves.
RE: In Poland, a typical unit of prescribing and dispensing are drug packs, e.g. 1 prescription could be issued for 1, 2 or much more packs of ‘Metformin Company X’ preparations holding, say, 30 tablets 1000 mg each. Thus, the number of dispensed packs is usually much higher than the number of dispensed prescriptions. Under these circumstances, title of Table 2 is correct, as it refers to the dispensation, assessed with the use of a unit of ‘drug packs’.
- To my opinion, fig 3 is not necessary.
RE: there was no Figure No. 3 in this manuscript
- Discussion: I m not sure that this study can be considered as a study on effectiveness of generic substitution in Poland
RE: Overall, this is not a study of ‘effectiveness of generic substitution’. We believe that our results clearly prove poor ‘effectiveness of implementation of the mechanism of the generic substitution’, and thus we described that in the discussion. Following your advice, we made this explanation (in the first paragraph of the Discussion section) more explicit, ho help understanding of our ideas.
Reviewer 2 Report
I have a few minor comments only. This implies that I consider the paper can also be published as submitted originally.
Line 34/35. And how about Europe? Non-adherence rates may differ among cultures and regions.
Line 110: Is product interchangeability assessed from a general public health perspective or patient centric perspective? For example if a patient would need a tablet with a score line, which is not present on the lowest priced generic product, tjhan the lowest priced product would not be suitable for that patient. Would the patient nevertheless have to pay extra for receiving an appropriate generic which holds a score line?
Table 1: Consider adding a column co-payment per pack. Values 49.78 and 50.22 are close and it is important that people don’t value this minor difference as irrelevant.
Line 294 talks about suboptimal use of generic substation whilst line 161 indicates relevant values for the same medicinal product may differ per date. This raises two important questions that preferably are addressed in the discussion section. A) If prices can change per period, do the authors think that patients should always be given the lowest priced product meaning that every time patients would go to the pharmacy, they may receive a different generic trademark? B) If yes, could you expand on any data regarding how repeated generic-generic substitution may or may nog affect the risk for medication errors and swapping products ?
Author Response
Dear Reviewer No. 2,
Thank you for your thorough review, and your positive opinion about our manuscript. Here we come with clear response to the points raised by you, as well as some additional explanations.
- Line 34/35. And how about Europe? Non-adherence rates may differ among cultures and regions.
RE: a clear reference to European (Swedish) scenario is provided
- Line 110: Is product interchangeability assessed from a general public health perspective or patient centric perspective? For example if a patient would need a tablet with a score line, which is not present on the lowest priced generic product, than the lowest priced product would not be suitable for that patient. Would the patient nevertheless have to pay extra for receiving an appropriate generic which holds a score line?
RE: The regulations governing Polish reimbursement system allow the patient to obtain the other version of the drug than the one prescribed by the physicians. Therefore, the patients may ask the pharmacist to dispense tablets of specific characteristics, such as small, big, round, long, capsules instead of tablets etc. etc. Any time, this may incur a diverse co-payment; some of these options would lead to higher co-payment. However, patients’ preferences differ, both across the patients, and within the patients, depending on the clinical scenario. Thus, the choice of patients’ reference may either increase, or decrease their co-payments, depending on what was put by the prescriber on the original prescription.
- Table 1: Consider adding a column co-payment per pack. Values 49.78 and 50.22 are close and it is important that people don’t value this minor difference as irrelevant.
RE: table 1 is very busy now, and we did not want to make it even more busy. Therefore, relevant numbers have been provided in the text (the second paragraph of the Results section), these being 3.86 vs 9.12 PLN per pack, respectively.
- Line 294 talks about suboptimal use of generic substation whilst line 161 indicates relevant values for the same medicinal product may differ per date. This raises two important questions that preferably are addressed in the discussion section. A) If prices can change per period, do the authors think that patients should always be given the lowest priced product meaning that every time patients would go to the pharmacy, they may receive a different generic trademark? B) If yes, could you expand on any data regarding how repeated generic-generic substitution may or may nog affect the risk for medication errors and swapping products ?
RE: Indeed, this question is already tackled in the Introduction and the Discussion section, for more details, see the reference #14
Reviewer 3 Report
The potential to improve medication adherence and reduce the public payer spending in Poland through an optimised implementation of the generic substitution: the win-win scenario suggested by the real-world Big Data analysis.- the title seems need to be more concise
Abstract
The aim of this study was to assess the effectiveness of implementation of this mechanism in Poland.
=Question: pls state what type of mechanism
This was a retrospective analysis of nationwide real-world Big Data corresponding to dispensation of Metformin preparations in 2019 in Poland. Relevant prescription and dispensation data were compared to assess the prevalence of generic substitution, and its economic consequences.
Dispensation=this is not the right term probably
The abstract should also describe how the data were analysed.
The term big data , I wonder if it is appropriate considering that Big Data is a collection of data that is huge in volume, yet growing exponentially with time. It is a data with so large size and complexity that none of traditional data management tools can store it or process it efficiently. Big data is also a data but with huge size.
This is especially obvious that under 2.1. Data and study design , the data analytic method or software was not described.
Intuitively, we know that generic substitution save cost. I was wondering if a revisit is necessary?
This especially points to the given conclusion that “ this suboptimal use of generic substitution was a reason for large lost savings in both patients and national payer. Thus, our findings point at the need to optimise implementation of generic substitution in Poland”
Author Response
Dear Reviewer No. 3,
Thank you very much for your thorough review. Here we come with clear response to the points raised by you, as well as some additional explanations.
- The potential to improve medication adherence and reduce the public payer spending in Poland through an optimised implementation of the generic substitution: the win-win scenario suggested by the real-world Big Data analysis.- the title seems need to be more concise
RE: Following your advice, the title was simplified a little to better reflect the results
- Abstract: The aim of this study was to assess the effectiveness of implementation of this mechanism in Poland. =Question: pls state what type of mechanism
RE: To make the sentence clearer, ‘this mechanism’ was replaced by ‘generic substitution mechanism’
- This was a retrospective analysis of nationwide real-world Big Data corresponding to dispensation of Metformin preparations in 2019 in Poland. Relevant prescription and dispensation data were compared to assess the prevalence of generic substitution, and its economic consequences.
- Dispensation=this is not the right term probably
RE: We believe that with more than 11,000 hits in PubMed database, the term of ‘drug dispensation’ seems to be often used in this context
- The abstract should also describe how the data were analysed.
RE: We used our best to describe the range and the methods of the data analysis in the abstract. Unfortunately, very low word limit does not allow for making this explanation more detailed.
- The term big data , I wonder if it is appropriate considering that Big Data is a collection of data that is huge in volume, yet growing exponentially with time. It is a data with so large size and complexity that none of traditional data management tools can store it or process it efficiently. Big data is also a data but with huge size.
RE: We made a search to find out whether any consensus definition is applicable to the term of ‘Big Data’, and its results were negative. E.g. according to Wikipedia, Big data is simple ‘data sets that are too large or complex to be dealt with by traditional data-processing application software’. In such a case, we believed we were authorized to use this term in case of the data set that we analysed.
- This is especially obvious that under 2.1. Data and study design , the data analytic method or software was not described.
RE: Overall, it was a custom-made analysis. We have added this information to the methods section.
- Intuitively, we know that generic substitution save cost. I was wondering if a revisit is necessary?
RE: In fact, the only justification of generic substitution is to save costs. What we had studied in fact was how well this rule is used in Polish healthcare system.
- This especially points to the given conclusion that “ this suboptimal use of generic substitution was a reason for large lost savings in both patients and national payer. Thus, our findings point at the need to optimise implementation of generic substitution in Poland”
RE: Let us repeat the message in the rephrased way: we found implementation of generic substitution mechanism in Poland far from optimal. These leads to huge losses in both patients and the public payer. Therefore, corrective actions should be taken.
Round 2
Reviewer 1 Report
The Fig2 is not necessary
Reviewer 3 Report
All comments have been adequately addressed.